

# $H_2O_2$ promotes trimming-induced tillering by regulating energy supply and redox status in bermudagrass

Shuang Li[*], Yanling Yin[*], Jianmin Chen, Xinyu Cui and Jinmin Fu

Coastal Salinity Tolerant Grass Engineering and Technology Research Center, Ludong University, Yantai, China

[*] These authors contributed equally to this work.

## ABSTRACT

Tillering/branching pattern plays a significant role in determining the structure and diversity of grass, and trimming has been found to induce tillering in turfgrass. Recently, it has been reported that hydrogen peroxide ($H_2O_2$) regulates axillary bud development. However, the role of $H_2O_2$ in trimming-induced tillering in bermudagrass, a kind of turfgrass, remains unclear. Our study unveils the significant impact of trimming on promoting the sprouting and growth of tiller buds in stolon nodes, along with an increase in the number of tillers in the main stem. This effect is accompanied by spatial-temporal changes in cytokinin and sucrose content, as well as relevant gene expression in axillary buds. In addition, the partial trimming of new-born tillers results in an increase in sucrose and starch reserves in their leaves, which can be attributed to the enhanced photosynthesis capacity. Importantly, trimming promotes a rapid $H_2O_2$ burst in the leaves of new-born tillers and axillary stolon buds. Furthermore, exogenous application of $H_2O_2$ significantly increases the number of tillers after trimming by affecting the expression of cytokinin-related genes, bolstering photosynthesis potential, energy reserves and antioxidant enzyme activity. Taken together, these results indicate that both endogenous production and exogenous addition of $H_2O_2$ enhance the inductive effects of trimming on the tillering process in bermudagrass, thus helping boost energy supply and maintain the redox state in newly formed tillers.

## INTRODUCTION

After cultivating a lawn, it typically demands significant human and material resources to enhance its visual appeal and extend its lifespan. Maintenance measures must align with the corresponding technical requirements. Among these measures, trimming holds a pivotal role in lawn maintenance (*Gu et al., 2015*). Regular and proper trimming can suppress the apical dominance of turfgrass, maintain a level lawn, foster grass branching, and reduce the competitiveness of weeds with higher elongation points (*Busey, 2003*; *DeBels et al., 2012*). Tillering phenotype is closely associated with energy utilization efficiency and a plants' adaptability to the environment, which, in turn, can enhance the resistance, adaptability,

Corresponding author
Jinmin Fu, turfcn@qq.com

and overall utilization value of plants (*Mason et al., 2014*). However, the effects of trimming on the tillering process remain largely unexplored and unclear.

Tillering or branching serves as a key determinant of plant stature, and its regulation is a significant manifestation of developmental plasticity in plant stature. This plasticity is closely linked to a plant's ability to adapt to the environment and directly influences its competitiveness for survival (*Leyser, 2003*). The process of tillering begins with the emergence of an axillary bud, originating from the axillary meristem located at the leaf axil. This bud undergoes two developmental stages, initiation and elongation, before maturing into a tiller. The development of axillary buds is a complex process intricately governed by external environmental factors, internal hormone levels, energy status, and gene expression (*Kebrom, 2017*). Apical dominance is a well-recognized phenomenon in which the growth of lower axillary buds is inhibited by the actively growing tip of the main stem. Decapitation experiments have provided evidence that removing the growing tip activates the lower axillary buds, resulting in their continuous growth and the subsequent formation of new lateral branches. This process is closely intertwined with hormonal regulation (*Barbier, Dun & Beveridge, 2017*). The strict polar transport of auxin dictates that it cannot directly enter the axillary bud, but rather indirectly inhibits axillary bud development through other signaling components (*Bennett et al., 2016*). Among these signals, cytokinin (CKs) exerts its effects by directly entering the bud and promoting bud outgrowth (*Müller & Leyser, 2011*). Exogenous application of CKs locally stimulates bud outgrowth (*Dun et al., 2012*) and increases CKs biosynthesis in stems and axillary buds during axillary bud outgrowth (*Tanaka et al., 2006*). Meanwhile, it has been shown that foliar application of CK-derived compounds improved the number of productive tillers and the grain yield in winter wheat and spring barley (*Koprna et al., 2021*).

Tiller initiation and elongation is an energy-consuming process. According to the nutritional hypothesis, the competition for sugars between the apical meristem and axillary meristem plays a vital role in maintaining apical dominance (*Barbier et al., 2019*). Previous studies have shown that moderate exogenous sucrose treatment significantly accelerate the initiation and sustained growth of axillary buds in several species, including Arabidopsis, rose, and pea (*Barbier et al., 2015*; *Mason et al., 2014*). Recent studies have demonstrated the importance of sugar as a crucial signaling molecule that affects bud growth (*Lastdrager, Hanson & Smeekens, 2014*). Sucrose has been identified as a long-distance signal that regulates axillary bud development by inducing axillary bud release prior to auxin action. Additionally, elevated sucrose levels inhibit the expression of *BRANCHED1* (*BRC1*)/*TEOSINTE BRANCHED1* (*TB1*), a key transcription factor in preserving bud dormancy (*Mason et al., 2014*).

Reactive oxygen species (ROS) comprise a class of oxygen-containing radicals, such as superoxide ($O_2^{\cdot-}$) and hydrogen peroxide ($H_2O_2$). They serve as ubiquitous signaling molecules and play a crucial role in various vital physiological processes in plants (*Ivanchenko et al., 2013*). In recent years, studies have demonstrated that ROS plays a vital role in regulating plant cell growth, directly or indirectly affecting plant growth and development. For example, ROS have been found to be involved in the elongation of root cells (*Dunand, Crèvecoeur & Penel, 2007*), seed germination (*Müller et al., 2009*)

and pollen tube development (*Potocký et al., 2007*). Among ROS molecules, $H_2O_2$ has a longer half-life and can function as a long-distance signaling molecule (*Mhamdi & Van Breusegem, 2018*). Some studies have shown a correlation between $H_2O_2$ metabolism and bud growth. Respiratory burst oxidase homolog (*rboh*) mutant plants displayed a highly branched phenotype, while the application of $H_2O_2$ suggested a negative effect of $H_2O_2$ on bud outgrowth (*Sagi et al., 2004*; *Chen et al., 2016*). In rosebushes, quiescent buds are prevented from outgrowth due to high levels of $H_2O_2$ (*Porcher et al., 2020*). However, it has been demonstrated that both clipping and grazing can induce $H_2O_2$ production and gene expression related to antioxidant pathways in the herbaceous perennial sheepgrass (*Huang et al., 2014*). Nevertheless, the regulation of regeneration and tiller processes following grass construction and the associated mechanisms involving $H_2O_2$ remain unclear.

Bermudagrass (*Cynodon dactylon* (L.) Pers.) is a widely used warm-season turfgrass that also serves as excellent ground cover and greenery due to its fast planting and strong stress resistance (*Beard, 2002*). At present, most bermudagrass varieties are established through asexual propagation, which heavily relies on the continuous production of tillers from stolon nodes, aboveground stems, and underground stem nodes. To optimize the quality of bermudagrass lawns and alleviate pressure on turfgrass resources, it is recommended to trim the grass to maximize its tiller capacity. This practice helps maintain the desired height and increase the density of bermudagrass lawns. However, the physiological and molecular mechanisms underlying the regulation of bermudagrass tillering by trimming remain unclear, particularly concerning the involvement of $H_2O_2$ in this process. In this study, we aim to characterize the spatio-temporal distribution changes of two well-established regulators, sugar and cytokinin, in response to trimming, and to investigate the potential involvement of $H_2O_2$ in trimming-induced tillering.

## MATERIALS AND METHODS

Portions of this text were previously published as part of a preprint (https://www.researchsquare.com/article/rs-3133950/v1)

### Plant material and growth conditions

The experimental material used was bermudagrass 'A12359' (2n = 36), which was planted in 30 pots (diameter: 15.5 cm, height: 17.5 cm) on November 3, 2021, and cultivated in the greenhouse of Binhai Grass Germplasm Resources Breeding Base located in the northern area of Ludong University, Yantai City, Shandong Province (121°36′N, 37°53′E). It received watering every two days and was provided with fresh 1/2-strength Hoagland's nutrient solution (formula: $NH_4H_2PO_4$ (0.5 mM), $KNO_3$ (2.5 mM), Ca $(NO_3)_2 \cdot 4H_2O$ (2.5 mM), $MgSO_4 \cdot 7H_2O$ (1 mM), $H_3BO_3$ (23 μM), $ZnSO_4 \cdot 7H_2O$ (0.38 μM), $CuSO_4 \cdot 5H_2O$ (0.16 μM), $MnCl_2 \cdot 4H_2O$ (4.5 μM), $H_2MoO_4$ (0.2 μM), Fe-EDTA (25 μM).) once a week to ensure adequate supply of water and nutrients. After the bermudagrass had developed abundant stolons, it was decapitated.

### Trimming treatment

The experimental materials were cultivated to have two tillers and then trimmed. During each treatment, bermudagrass was uniformly trimmed using auxiliary tools to maintain

a stubble height of 5 cm. Both control and treatment groups consisted of six biological replicates. Tiller buds containing two leaves and one central bud were defined as tillers. From the beginning of the experiment until 10 days after treatment, the number of tillers was counted every 2 h. At 2, 4, 6, 8 and 10 h after treatment, the number of tillers on the main stem of bermudagrass in both the control and treatment groups were counted multiple times, and the dynamic change curve of the number of tillers in response to grass trimming was plotted. When bermudagrass produces more stolons, the stolons with uniform growth were selected and labeled accordingly. Two groups, control and trimming, each consisting of five stolons, were established. Trimming was simulated by removing the topmost stem node. From the beginning of the experiment to 6 days after treatment, the length of tiller buds at different nodes was measured daily. Subsequently, tiller buds at the second node of the control and trimming groups were compared to analyze the pattern of their dynamic changes in response to trimming.

## Exogenous treatment

Three pots of bermudagrass exhibiting similar growth were divided into three groups: one group was sprayed with 50 $\mu$M 6-Benzylaminopurine (6-BA), the second group was sprayed with 200 mM sucrose, and the third group was sprayed with distilled water as a control. To measure the length of tiller buds at the end of the treatment, six stolons were selected for each treatment. The treatments were performed every other day for a total of six times.

Four treatment levels were established for bermudagrass, including a non-trimming group consisting of a control and exogenous $H_2O_2$, and a trimming group consisting of a control and exogenous $H_2O_2$, with six replicates per treatment level. Bermudagrass with a consistent initial tiller count of four was selected. The leaves were treated with 2 mM $H_2O_2$ until they were dripping, and this process was repeated every other day (concentrations were determined by pre-experimentation). The tillers were counted after each treatment until the 8th day.

## Gene expression analysis

From each of the first four nodes of bermudagrass stolons, 0.1 g was taken after 6 h of trimming treatment. 0.1 g of nodal sample was taken after 0, 1, 3, 6, 12, and 24 h upon trimming treatment. Additionally, 0.1 g of nodal sample was taken after exogenous $H_2O_2$, including four treatment levels. Three replicates of each sample were ground to powder in liquid nitrogen. Total RNA was extracted using the Plant RNA Kit with genomic DNA enzyme (Vazyme Biotech Co., Ltd, Jiangsu, China), and the RNA concentration and quality were measured using a NanoDrop microspectrophotometer (Eppendorf, Barkhausenweg, Germany). Subsequently, 5 $\mu$L of RNA was reverse transcribed to obtain cDNA with HiScript II 1st Strand cDNA Synthesis Kit with genomic DNA enzyme (Vazyme Biotech Co., Ltd, Jiangsu, China), which was then employed as a template for qRT-PCR (Quantitative real-time reverse transcription-PCR, Thermo Fisher Scientific, Waltham, MA, USA) to detect gene expression using SYBR qPCR Master Mix (Vazyme Biotech Co., Ltd, Jiangsu, China).

## Zeatin and sugar measurement

After 24 h of trimming, nodes were collected and freeze-dried in liquid nitrogen, then stored at $-80$ °C until needed. The dried plant materials were then homogenized and powdered in a mill.

For zeatin detection, 100 mg of dried powder was extracted with 1.5 mL of a mixed solution of MeOH: $H_2O$: FA (79.9: 20: 0.1). The extract was vortexed and subjected to ultrasound for 30 min, then stored at 4 °C for 12 h. After centrifugation, supernatants were collected. The residue was reextracted with one mL of MeOH under ultrasound for 30 min and centrifuged. The supernatants were combined, evaporated to dryness under a nitrogen gas stream, and reconstituted in 100 μL of a mixed solution of MeOH: $H_2O$ (50: 50).

For sugar detection, 30 mg of dried powder was extracted with a mixed solution of Ethanol: $H_2O$ (80: 20) in a volume of 700 μL. The extract was vortexed and subjected to ultrasound for 30 min, placed in a 70 °C water bath, and kept there for 2 h. Next, 700 μL of chloroform was added and supernatants were collected after centrifugation. The supernatants were then evaporated to dryness under a nitrogen gas stream and reconstituted in a mixed solution of ACN: $H_2O$ (75: 25) in a volume of 100 μL.

Finally, the solution was filtered through a 0.22 μm filter for further LC-MS analysis.

Phytohormones contents were detected by Guocangjian (http://www.targetcrop.com/) based on the Sciex 4500 LC-MS/MS platform.

## OJIP fluorescence transient test

After trimming the first node of bermudagrass stolon for 6 and 24 h, intact leaves (the two foremost leaves of the stolon) and the leaves of newly formed tillers were collected in each treatment, which was repeated six times. The samples to be tested were subjected to the dark treatment for 30 min before being measured by a chlorophyll fluorimeter (PAM2500). The leaves of $H_2O_2$ exogenous treatment at the four treatment levels were also measured using a chlorophyll fluorometer (PAM2500) to obtain OJIP fluorescence transient curves.

## Starch content determination

After trimming the stem nodes for 0 and 24 h, the leaves of newly formed tillers were sampled, dried, and weighed (0.5 g). Each treatment was replicated three times. A total of 6–7 ml of 80% ethanol was added to the sample, followed by extraction in a water bath maintained at 80 °C for 30 min before centrifugation. This extraction process was repeated twice, and the precipitate was collected. Next, 3 ml of distilled water was added to the precipitate, which was then boiled in a water bath for 15 min. After cooling, 2 ml of chilled 9.2 M perchloric acid was added, and the sample was extracted for 15 min and centrifuged. The precipitate was then mixed with 4.6 M perchloric acid for 15 min and centrifuged. The precipitate was washed twice, and combined with centrifugal liquid to make a constant volume of 50 ml. A total of 1 ml of the extract and 5 ml of anthrone were mixed and shaken well in a boiling water for 10 min. The mixture was then cooled and the wavelength was measured at 625 nm with a spectrophotometer to calculate the starch content (*Ahamed et al., 1996*).

## H$_2$O$_2$ content determination

After being subjected to trimming treatment for 0, 1, 3, 6, 12, 24, 48 and 72 h, the sample was weighed at 0.1 g. Subsequently, nine times the volume of PBS (pH 7.0–7.4, 0.1 mol/L) was added following a weight (g) to volume (mL) ratio of 1:9. The supernatant homogenate was mechanically shaken under an ice-water bath and collected for determination using a hydrogen peroxide (H$_2$O$_2$) test kit (Jiancheng, Nanjing, China).

## Antioxidant enzyme activity assay

After treatment, 0.25 g of the sample was ground in liquid nitrogen. Subsequently, 4 ml of phosphate buffer (150 mM, pH =7.0) was added, and the mixture was centrifuged at 12,000 r/min and 4 °C for 20 min. The resulting supernatant was considered as the crude enzyme extract.

The activity of superoxide dismutase (SOD) was measured according to the method previously described, which was based on the reduction of its inhibition to nitro blue tetrazolium (NBT) (*Dhindsa, Plumb-Dhindsa & Thorpe, 1981*). For peroxidase (POD) activity assay, we followed the procedure outlined by *Pütter & Becker (1983)*. Catalase (CAT) activity was determined essentially according to the method described by EI-Moshaty (*El-Moshaty et al., 1993*). The decrease in H$_2$O$_2$ was monitored at 240 nm and quantified using the molar extinction coefficient of 36 M$^{-1}$ cm$^{-1}$.

# RESULTS

## Trimming induces tillering in bermudagrass

To determine the effect of trimming on tillering in bermudagrass, we first measured the tillering process in the main stem and stolon node in response to trimming. Two days after trimming, a significant difference in tiller count was observed in the main stem compared to the control group, and this difference became more pronounced as the treatment period extended. On the 10th day of treatment, the control group had nine tillers while the trimming group had 13 tillers, representing a 44% increase in tiller generation resulting from trimming (Fig. 1B). In addition, trimming stimulated the outgrowth of tiller buds located at the morphologically upper 1st, 2nd, and 3rd nodes of the stolon, with the most significant effect observed at the 1st node. However, trimming could not promote the growth of tiller buds at the 4th node (Figs. 1C and 1D).

We further analyzed the dynamic change pattern of the 2nd buds in response to trimming. It was found that on the second day after treatment, tiller buds appeared longer compared to the control, and the difference gradually increased with the extension of treatment time. On the 6th day of treatment, the length of the tiller buds that underwent trimming treatment was 3.2 times longer than the control (Figs. 1E and 1F). Throughout the treatment, there was no significant change in the length of tiller buds in intact stolons, while the length of tiller buds showed an S-shaped growth pattern after trimming. These results indicated that the sprouting and growth of tiller buds occurred earlier in the trimming group than in the control group.

After trimming treatment, the marker gene *TEOSINTE BRANCHED1* (*TB1*), which inhibits bud outgrowth, was down-regulated after 6 h and reached its lowest expression

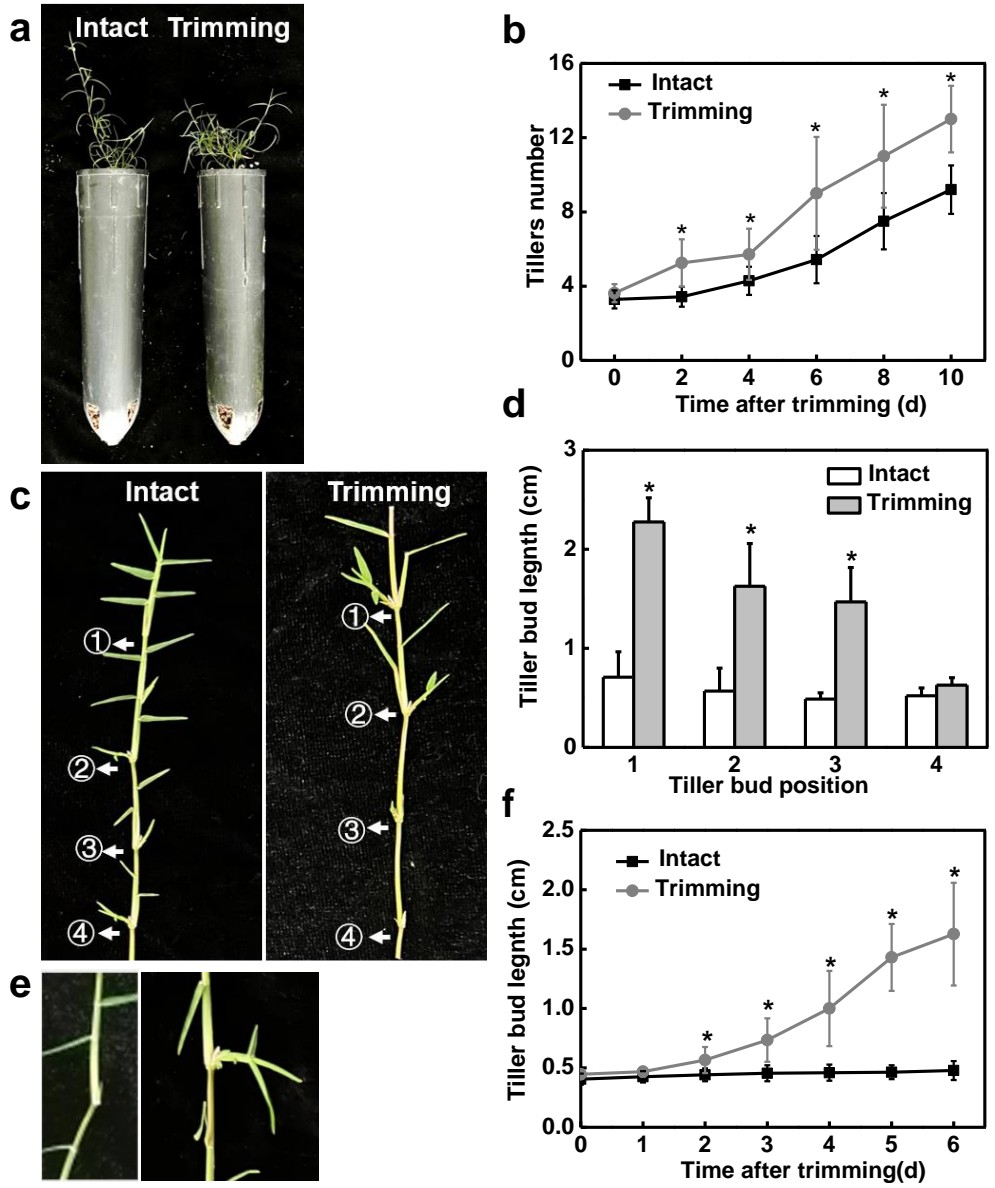

**Figure 1** **Trimming induces tillering in bermudagrass.** (A) Phenotype of the effect of trimming on the main stem tillering. (B) Response of main stem tiller count to changes in trimming time, with asterisks indicating significant differences between intact and trimmed samples by $t$-test ($P < 0.05$). (C) Phenotype of the effect of trimming on the stolon tillering. (D) Changes in length of stolon tiller buds at the 1st, 2nd, 3rd, and 4th nodes after 6 d of trimming treatment, with asterisks indicating significant differences between intact and trimmed samples by $t$-test ($P < 0.05$). (E) Phenotype of the change in the length of tiller buds at the 2nd node. (F) Response of tiller bud length at the 2nd node to changes in trimming time, with asterisks indicating significant differences between intact and trimmed samples by $t$-test ($P < 0.05$).

level at 12 h, indicating a consistent effect of trimming on stolon tiller bud induction (Fig. 2A). Regarding organ-specific expression, *TB1* was significantly down-regulated at the 2nd and 3rd nodes after 6 h of trimming compared to the control (Fig. 2B).

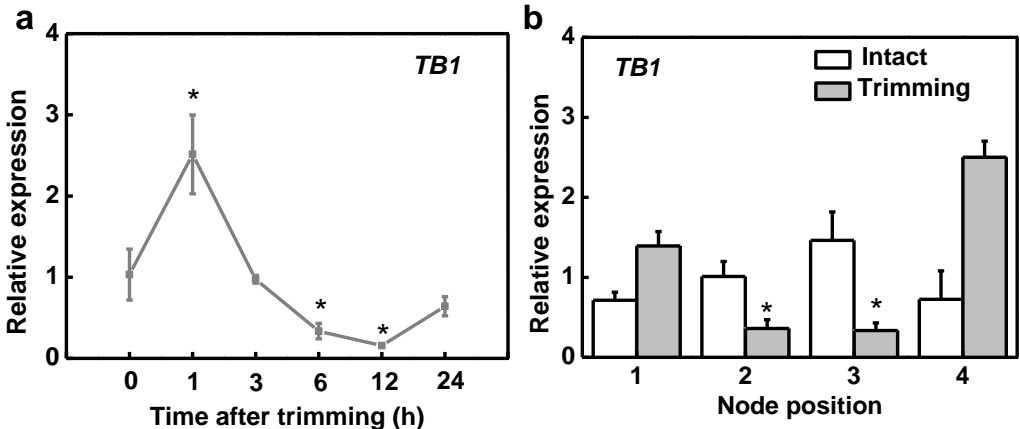

**Figure 2** *TB1* expression dynamics after trimming in stem nodes of bermudagrass. (A) Changes in *TB1* expression over time of trimming, with asterisks indicating significant difference between each trimming time and 0 h by *t*-test (*P* < 0.05). (B) Changes in *TB1* expression at each stem node after 6 h of trimming, with asterisks indicating significant differences between intact and trimmed samples by *t*-test (*P* < 0.05).

## Trimming induced temporal and spatial changes in CKs biosynthesis

Considering that CKs is induced by decapitation and promotes bud outgrowth, we assessed the changes in CKs biosynthesis in response to trimming. The active form of CKs zeatin (tZ) accumulated significantly only at the 1st node of the stolon (Fig. 3A). The spatial expression pattern of the genes related to CKs biosynthesis showed that *LONELY GUY1* (*LOG1*) and *ISOPENTYL TRANSFERASE1* (*IPT1*) and were significantly up-regulated at the 1st and 2nd nodes 6 h after trimming treatment (Figs. 3B and 3C). We identified 15 cytokinin oxidase/dehydrogenase (*CKX*) genes in the bermudagrass genome and examined their temporal response to trimming. Subsequently, we screened the two most promising genes, *CKX10* and *CKX12* (Fig. S1). Our results revealed a significant down-regulation of *CKX10* and *CKX12* at the 1st-3rd nodes following a 6 h trimming treatment (Figs. 3D and 3E). Furthermore, the time-course expression of these genes at the 1st node showed that *LOG1* was rapidly up-regulated by 3.1 times 1 h after trimming treatment and maintained this expression level until 12 h after trimming (Fig. 3F). *IPT1* was slightly induced at the early stage of treatment and peaked at 12 h after treatment, showing a 273-fold increase (Fig. 3G). Meanwhile, *CKX10* and *CKX12* exhibited a rapid down-regulation of 0.66-fold and 0.27-fold, respectively, at 1 h after trimming, followed by a sustained downward trend. (Figs. 4G and 4I). Therefore, our results indicated that trimming stimulated the accumulation of CKs, which was dependent on the node position.

Next, we investigated the effects of exogenously applied 6-BA, a synthetic cytokinin, on tiller bud induction. The results demonstrated that the exogenous application of 6-BA effectively induced tiller buds at the 2nd, 3rd, 4th, and 5th nodes (Fig. S2).

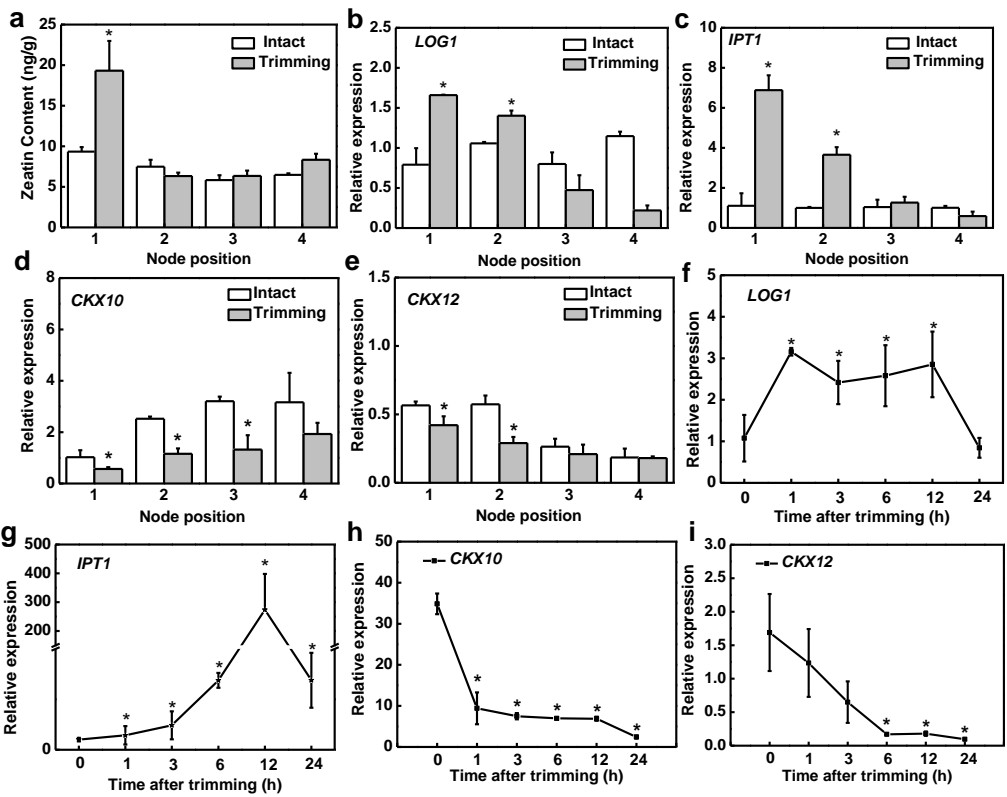

**Figure 3** **Altered CKs and cytokinin-related genes in stem nodes of bermudagrass after trimming.** (A) Zeatin content at 1–4 nodes 24 h after trimming. (B, C) Changes in *IPT1* and *LOG1* at 1–4 nodes 6 h after trimming, with asterisks indicating significant differences between intact and trimmed samples by *t*-test ($P < 0.05$). (D, E) Changes in *CKX10* and *CKX12* at 1–4 nodes 6 h after trimming, with asterisks indicating significant differences between intact and trimmed samples by *t*-test ($P < 0.05$). (F, G) Changes in *IPT1* and *LOG1* over time of trimming, with asterisks indicating significant differences between each trimming time and 0 h by *t*-test ($P < 0.05$). (H, I) Changes in *CKX10* and *CKX12* over time of trimming, with asterisks indicating significant differences between each trimming time and 0 h by *t*-test ($P < 0.05$).

## Trimming increased the energy allocation towards the tillering process by enhancing photosynthesis

Since tillering is an energy-consuming process, we sought to investigate whether photosynthesis played a role in the response to trimming by measuring various photosynthesis-related parameters. Leaves of plants show a characteristic polyphasic Chl a fluorescence induction curve under high intensity continuous light illumination, termed as the OJIPSMT transient. The OJIP fluorescence transient curve exhibited an elevation at both 6 and 24 h after trimming, with a more pronounced enhancement observed at 24 h compared to 6 h (Fig. 4A). To further study the effect of trimming on the photosynthetic system, we derived additional parameters through the JIP test (Table S2). It is evident that some parameters were enhanced after treatment. Notably, basic photosynthetic parameter (Mo), quantum yield parameter ($\varphi$Eo) and efficiency parameter ($\Psi$o), as well as performance index $PI_{cs}$ ($P<0.01$) showed highly significant differences after 6 h of

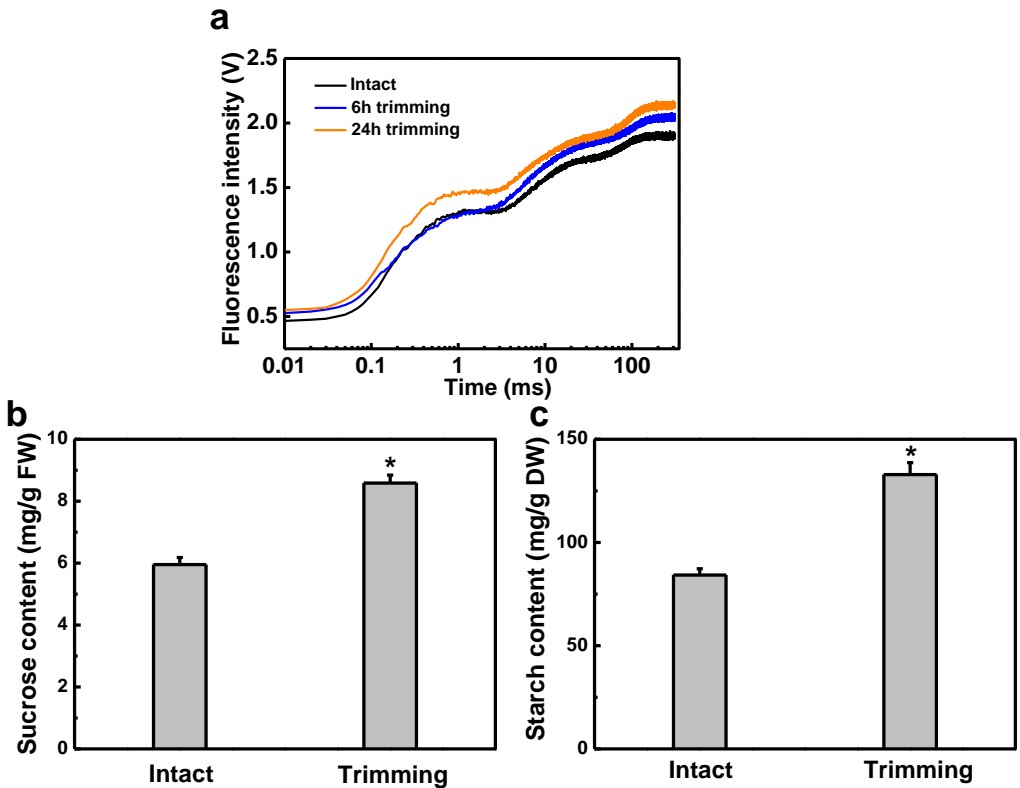

**Figure 4** **Effect of trimming on photosynthesis and sugar content in bermudagrass.** (A) Effects of trimming on OJIP fluorescence transients in newly formed tiller leaves. (B) Changes in sucrose content in newly-formed tiller leaves after 24 h of trimming. (C) Changes in starch content in newly-formed tiller leaves after 24 h of trimming. An asterisk (*) indicates significant differences between intact and trimmed samples, as determined by $t$-test ($P < 0.05$).

trimming. The performance indices $PI_{ABC}$, $PI_{Total}$, and $PI_{cs}$, which provide a more accurate reflection of the photosynthetic state, also increased after trimming as seen in Table S2.

Sucrose is the end product of photosynthesis, and starch serves as the primary storage form of photosynthates. In line with the increased potential for photosynthesis, there was a 44.28% increase in sucrose content (Fig. 4B) and a 57.77% increase in starch content (Fig. 4C) in the leaves of newly formed tillers following trimming.

We further studied the effect of trimming on sugar levels in stolon nodes by analyzing both the sugar content and expression level of the sucrose biosynthesis gene, sucrose phosphate synthase (*SPS*). Our findings indicated that sucrose and glucose content significantly increased at the 3rd and 4th nodes (Figs. 5A and 5B), with fructose content notably higher at the 3rd node compared to the control (Fig. 5C). Moreover, *SPS* was significantly up-regulated at the 1st, 2nd, and 3rd nodes of the stolon compared to the control 6 h after trimming treatment (Fig. 5D). The time-course expression analysis of *SPS* at the 3rd node revealed that it was up-regulated starting from 3 h after trimming, reaching its peak at 12 h (Fig. 5E). Furthermore, exogenously spraying sucrose induced the formation of tiller buds (Fig. S2).

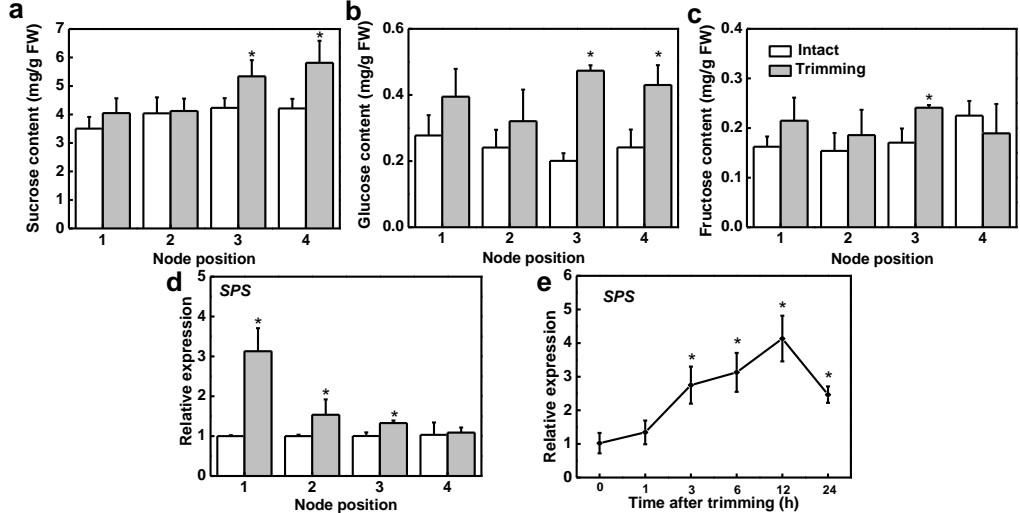

**Figure 5** **Changes in sugar content and its related synthetic genes in stem nodes of bermudagrass after trimming.** (A, B, C) Sucrose, glucose, and fructose content at 1–4 nodes 24 h after trimming, with an asterisk (*) indicating significant differences between intact and trimmed samples by $t$-test ($P < 0.05$). (D) Changes in *SPS* at 1–4 nodes 6 h after trimming, with an asterisk (*) indicating significant differences between intact and trimmed samples by $t$-test ($P < 0.05$). (E) Changes in *SPS* over time of trimming, with an asterisk (*) indicating significant differences between each trimming time and 0 h by $t$-test ($P < 0.05$).

## $H_2O_2$ facilitated trimming-induced tillering by enhancing photosynthesis

Given the role of $H_2O_2$ in controlling bud outgrowth, we examined its response to trimming and its effect on tillering. Our findings indicated that $H_2O_2$ levels in both the clipped leaves and newly-formed tillering leaves increased rapidly within 1 h of trimming and remained elevated for 24 h. At 48 h after treatment, $H_2O_2$ levels in the leaves of newborn tillers decreased to a level below that of the control and did not increase (Fig. 6A). Moreover, the exogenous application of $H_2O_2$ resulted in a 1.7-fold increase in the number of tillers in intact plants (Figs. 6B and 6C). Trimming stimulated tillering, which was further enhanced by the application of $H_2O_2$.

Chlorophyll fluorescence in the leaves of the newly-formed tillers was measured. The exogenous application of $H_2O_2$ led to an increase in the values represented by the OJIP curve for the leaves of untrimmed-plants, and the enhancing effects of trimming on the OJIP curve were further amplified by the application of $H_2O_2$ (Fig. 7A). Consistently, the application of exogenous $H_2O_2$ to intact plants resulted in a 1.48-fold increase in sucrose content, which was further increased by 1.55-fold upon trimming (Fig. 7B). Similarly, exogenous $H_2O_2$ increased starch content in intact plants, with the highest starch content observed in the trimming+$H_2O_2$ treatment group, showing a 1.41-fold increase compared to the trimming treatment group (Fig. 7C).

The dual role of $H_2O_2$ in regulating plant growth prompted us to assess the antioxidant capacity of plants treated with $H_2O_2$. Interestingly, the activities of SOD, POD, and CAT were enhanced by both trimming and exogenous $H_2O_2$ after 24 to 72 h of trimming

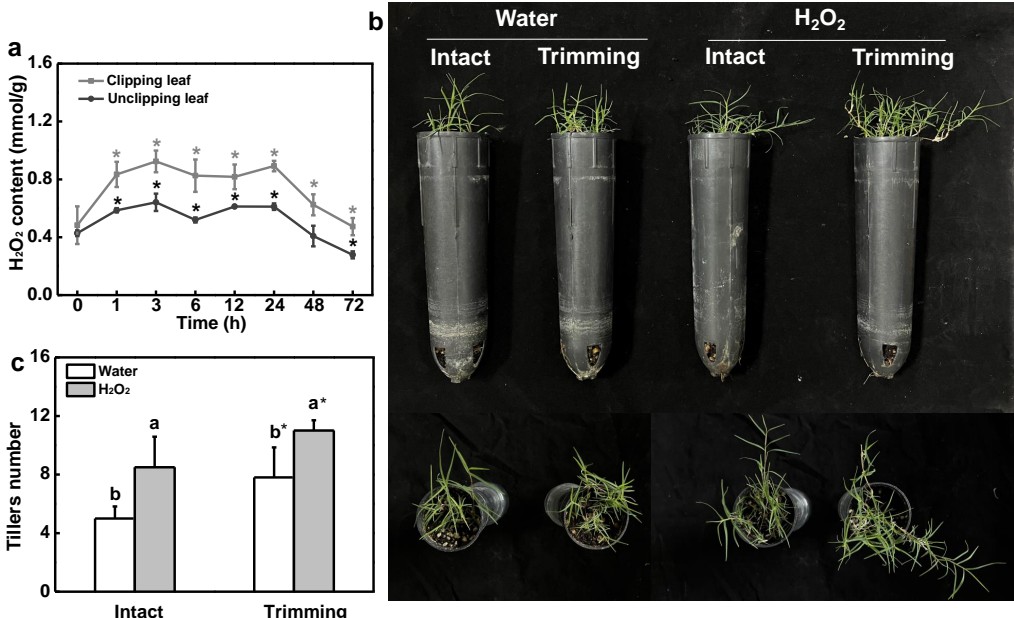

**Figure 6  Effect of H$_2$O$_2$ on tillering in bermudagrass.** (A) Response pattern of H$_2$O$_2$ content in clipped leaves and unclipped leaves over time of trimming. Clipped leaves represent wounded leaves, while unclipped leaves represent the leaves of newly formed tillers, with asterisks indicating significant differences between each trimming time and 0 h by $t$-test ($P < 0.05$). (B) Four treatment level phenotypes. (C) Changes in tiller count at four treatment levels. "a" and "b" indicate differences between different treatments of the same state, while asterisks (*) indicate differences between different states of the same treatment. $T$-test was used for differences ($P < 0.05$).

treatment. Importantly, exogenous H$_2$O$_2$ further enhanced the antioxidant enzymes activity triggered by trimming after 48 h of exogenous treatment (Figs. 8A–8C).

Our results showed that both CKs and H$_2$O$_2$ promoted bermudagrass tillering. Therefore we considered about interplay of CKs and H$_2$O$_2$ in regulation of tillering after trimming. Exogenous H$_2$O$_2$ promotes the up-regulation of CKs biosynthetic genes *IPT1* and *LOG1* in intact plants. The trimming+ H$_2$O$_2$ treatment exhibited the strongest up-regulation trend, being 1.39-fold and 0.96-fold higher than the trimming treatments, respectively (Figs. 9A and 9B). Regarding *CKX10* and *CKX12*, these genes were significantly down-regulated after exogenous H$_2$O$_2$ treatment in intact plants, with trimming+ H$_2$O$_2$ showing a 0.52-fold and 0.12-fold decrease compared to the trimming group (Figs. 9C and 9D).

# DISCUSSION

## Trimming triggered tillering in bermudagrass by inducing cytokinin

The development and growth of tillers have a significant impact on the overall growth of plants. Currently, decapitation has been demonstrated to induce tiller formation in many species such as pea, Arabidopsis, and rose (*Morris et al., 2005*; *Tatematsu et al., 2005*; *Mason et al., 2014*). However, there are fewer studies on the effect of decapitation on perennial herbs. In barley, simulated grazing through clipping revealed that a certain degree

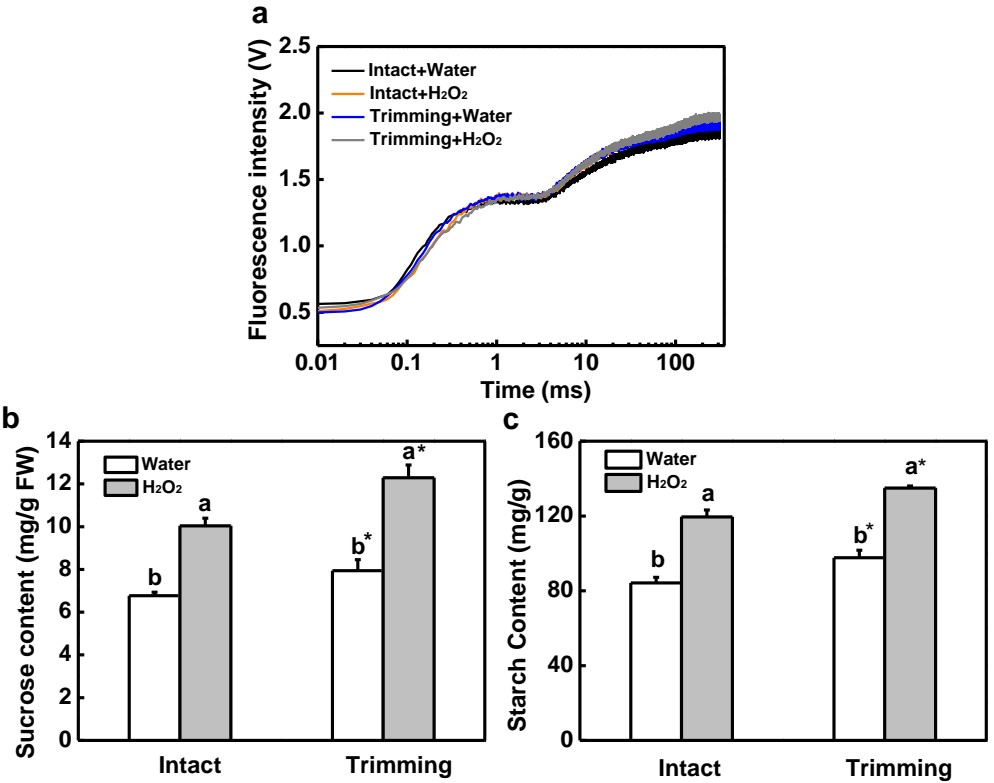

**Figure 7** **Effect of exogenous H₂O₂ on photosynthesis and sugar content in bermudagrass.** (A) OJIP fluorescence transient curve of the leaves of newly formed tillers under the four treatments. (B, C) Sucrose and starch content of newly-formed tiller leaves under the four treatments. "a" and "b" indicate differences between different treatments of the same state, while asterisks (*) indicate differences between different states of the same treatment. $T$-test was used for differences ($P < 0.05$).

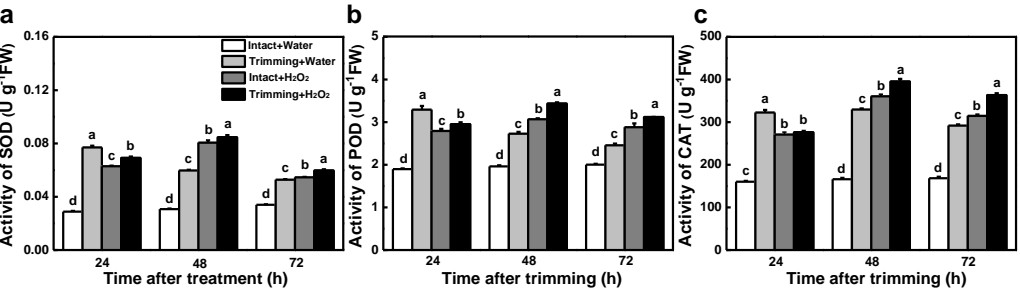

**Figure 8** **Effect of exogenous H₂O₂ on the antioxidant enzyme activity in bermudagrass.** (A, B, C) Trends of three antioxidant enzyme activities under four treatments, with letters a, b, c, and d indicating significant differences between the treatments. One-Way ANOVA was used for differences ($P < 0.05$).

of clipping could increase the number of tillers (*Yuan, Li & Yang, 2020*). This finding is consistent with our results, which indicate that trimming promotes tillering in the main stem and tiller bud development in the stolon node (Fig. 1).

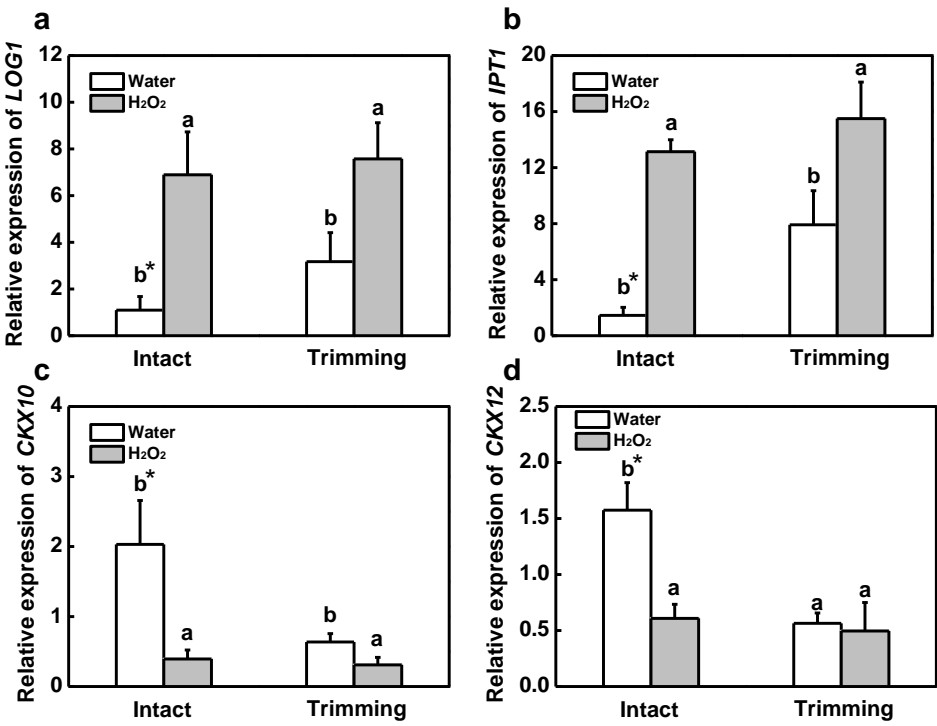

**Figure 9** **Changes in cytokinin-related gene expression after exogenous H₂O₂ treatment.** (A, B) *LOG1* and *IPT1* expression dynamics. (C, D) *CKX10* and *CKX12* expression dynamics. "a" and "b" indicate differences between different treatments of the same state, while asterisks (*) indicate differences between different states of the same treatment. *T*-test was used for differences ($P < 0.05$).

*BRC1/TB1* is considered a key transcription factor for inhibiting lateral bud outgrowth. Multiple pathways regulating branching and tillering converge to regulate the expression of *BRC1/TB1* (*Takeda et al., 2003*; *Minakuchi et al., 2010*). In a previous study conducted on bermudagrass, the transcriptional level of *BRC1/TB1* was found to be negatively correlated with tiller capacity (*Zhang & Liu, 2018*). We observed significant inhibition of this gene at the 2nd and 3rd nodes of bermudagrass stolon after trimming treatment (Fig. 2B). These results suggest that *TB1* may play a role in the trimming-induced tillering at specific node positions.

The development of tillers/branches and their formation process are regulated by interactions between multiple plant hormones. The accumulation of CKs in lateral buds and stems, as well as the expression of related genes, reflects the important role of CKs in regulating lateral bud outgrowth (*Young et al., 2014*). Rapid accumulation of CKs in stem nodes and axillary buds has also been observed in pea (*Cao et al., 2023*) and rose (*Roman et al., 2016*) after decapitation. Moreover, decapitation induces transient expression of CKs biosynthetic genes *IPT* and *LOG* (*Cao et al., 2023*; *Tanaka et al., 2006*). While deactivation of cytokinin is the sole responsibility of the enzyme called cytokinin oxidase/dehydrogenase, *CKX* (*Jiang et al., 2016*). Reducing *CKX* expression promotes the accumulation of CKs in organs, it demonstrated in rice (*Ashikari et al., 2005*) and

barley (*Zalewski et al., 2010*). Some studies also shown that exogenous application of CKs oxidase/dehydrogenase inhibitors in winter wheat and spring barley increased the CKs activity (*Nisler et al., 2021*). In our study, both CKs content and its biosynthetic genes were found to increase only at the 1st node after trimming. The *CKX* gene was significantly down-regulated at 1st to 3rd node (Fig. 3). This is consistent with findings in pea, where CKs levels were significantly enhanced at the upper nodes after decapitation (*Cao et al., 2023*). Therefore, trimming enhances CKs accumulation in bermudagrass stolons and the accumulation of CKs induced by trimming may depend on the distance from the site of trimming.

### Trimming enhanced energy supply during the tillering process

Regrowth and tillering after trimming require sufficient energy supply to the newly formed tillers (*Barbier et al., 2019*). We observed an improvement in the photosynthetic rate of bermudagrass after trimming, with photosynthetic products accumulating in the leaves (Fig. 4). This result suggests that trimming can promote the accumulation of energy metabolites for tiller production. Our finding is consistent with the results of studies on tea trees, where enhanced photosynthetic performance contributed to bud growth (*Yue, Wang & Yang, 2021*). Light and sucrose serve as signals and energy sources for bud growth, thereby regulating plant growth and development. Moreover, sugars (photosynthates) act as signaling molecules to induce axillary bud development (*Signorelli et al., 2018*). Decapitation is a means to rapidly induce sugar signal transduction and promote axillary bud release (*Mason et al., 2014*). Decapitation induces axillary bud development by reducing competition for sugars in the apical bud, leading to an accumulation of sugars in the axillary buds and thus triggering a series of pathways that induce axillary bud germination. In tobacco, the expression of sucrose biosynthesis genes, *SPS* and sucrose phosphate phosphatase (*SPP*), is up-regulated after topping (*Wang et al., 2018*). Similarly, after the topping of chrysanthemums, the expression levels of sucrose carriers (*SUCs*) and Sugars Will Eventually be Exported Transporters (*SWEETs*) increased (*Sun et al., 2021*). Meanwhile, the sucrose content increased significantly after trimming (*Mason et al., 2014*). Our results also showed significant up-regulation of the *SPS* gene and accumulation of sugar content at the 3rd and 4th nodes (Fig. 5), suggesting the potential role of sugar in trimming-induced bud development.

### $H_2O_2$ contributed to trimming-induced tillering by enhancing photosynthesis and antioxidant capacity

ROS, particularly $H_2O_2$, have been demonstrated to be important regulators of bud outgrowth. Studies have shown that $H_2O_2$ levels remain high in the dominant bud (*Chen et al., 2016*; *Porcher et al., 2020*). In rosebush, the $H_2O_2$ content was found to gradually decrease only in the buds and not in the neighboring stems after decapitation, suggesting that local changes in $H_2O_2$ content were not a result of a systematic response to wounding (*Porcher et al., 2020*). On the contrary, the $H_2O_2$ content showed a continuous increase in both the clipped leaves and the adjacent intact leaves in the newly formed tillers (Fig. 6A), suggests that trimming-induced tillering is accompanied by early $H_2O_2$ accumulation. This

result contradicts previous research and suggests that trimming may result in a wounding response in bermudagrass.

Exogenous application of $H_2O_2$ has been shown to severely reduce bud outgrowth in tomato and rosebush by increasing the $H_2O_2$ levels (*Chen et al., 2016*; *Porcher et al., 2020*). However, we obtained opposite results in our study, where exogenous $H_2O_2$ was found to promote tillering after trimming in bermudagrass (Figs. 6B and 6C). Nevertheless, the fact that bud outgrowth requires a reduced status led us to consider the effects of $H_2O_2$ on antioxidant capacity. Studies have shown that activation of the $H_2O_2$ scavenging system, such as an increase in glutathione (GSH) level, is linked to greater bud outgrowth. Furthermore, pretreatment with $H_2O_2$ has been shown to enhance antioxidant-related enzyme activity and improve stress resistance (*Guler & Pehlivan, 2016*; *Sathiyaraj et al., 2014*). Additionally, grazing in sheepgrass has also been shown to increase the expression of antioxidant-related genes (*Huang et al., 2014*). These findings indicate that early burst of $H_2O_2$ enhances the antioxidant enzyme activities induced by trimming, which is consistent with our results (Fig. 8). Additionally, we observed that hydrogen peroxide improved the photosynthetic properties of bermudagrass and increased the accumulation of energetic substances (Fig. 8). This aligns with previous studies that have shown exogenous $H_2O_2$'s ability to restore photosynthetic efficiency in tomato (*Nazir, Hussain & Fariduddin, 2019*) and increase sugar and starch content in melons (*Ozaki et al., 2009*). Moreover, $H_2O_2$ further upregulated the genes related to CKs biosynthesis and downregulated its degradational genes under trimming condition. These findings imply an integration between $H_2O_2$ and CKs in inducing tillering of bermudagrass. Taken together, it is possible that $H_2O_2$ acts as a second messenger, rather than an oxidized molecular, in response to trimming, helping to maintain the redox status and enhance energy supply during tiller growth. Our findings suggest that the role of $H_2O_2$ in controlling tillering in trimming-tolerant bermudagrass differs from its regulatory role in bud outgrowth in other species.

## CONCLUSION

Trimming has become an essential part of bermudagrass lawn management. Our study suggests that, similar to decapitation in other species, trimming-induced tillering is accompanied by the inhibition of *TB1* expression and the accumulation of CKs in nodes. Trimming promotes energy supply to the regrowth of newly formed tillers by enhancing photosynthesis. $H_2O_2$ responds rapidly to trimming, and exogenous application of $H_2O_2$ increases the number of tillers in bermudagrass. $H_2O_2$ not only enhances photosynthetic potential and energy reserves, but also primes antioxidant enzymes to maintain the redox status of newly formed tillers. Meanwhile, it affects the expression of CKs-related genes. Therefore, $H_2O_2$ appears to be a second messenger that relays trimming-related signals and stimulates regrowth after trimming (Fig. 10).

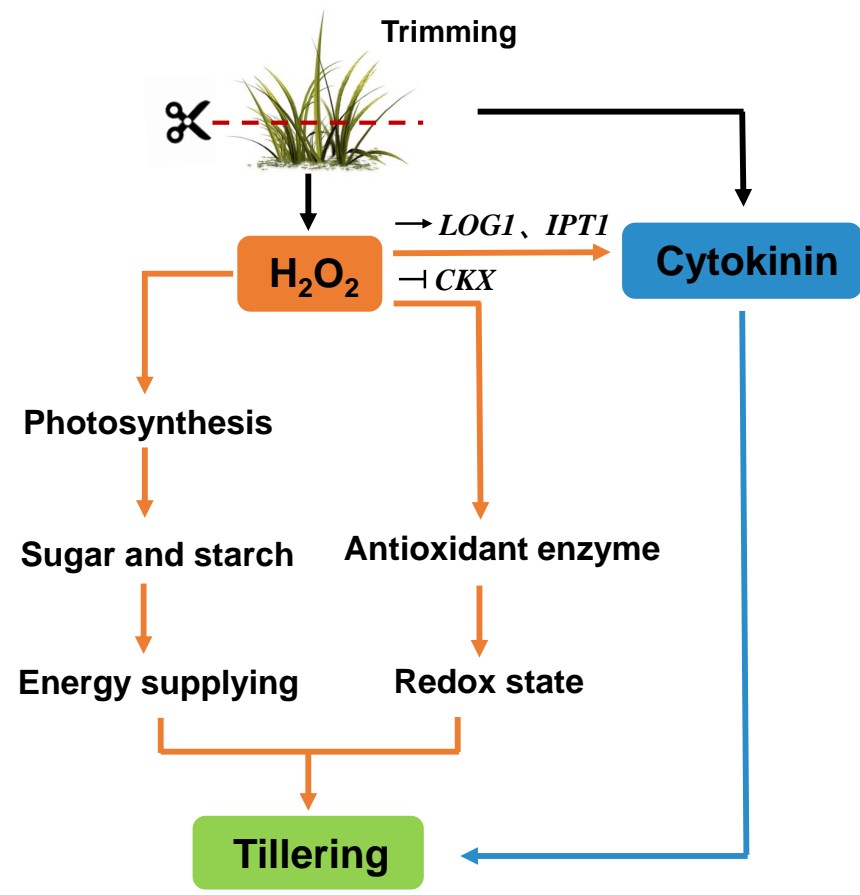

**Figure 10  Signaling pathway of trimming-induced tillering in bermudagrass.** Trimming induces CKs production in axillary buds, promoting their growth and activating the $H_2O_2$ mechanism. Moreover, this helps enhance the photosynthesis and accumulation of energetic material, while also increasing antioxidant enzyme activity to maintain the redox state of bermudagrass and promote tiller formation. It also can stimulate the up-regulation of CKs biosynthetic genes *IPT* and *LOG*, inhibited the expression of the cytokinin catabolic gene *CKX*, which affected the tillering state of bermudagrass after trimming.

## ACKNOWLEDGEMENTS

The authors thank all teachers for their help and guidance in the experiment, and my classmates for their guidance and help in learning and life.

### Funding

This work was supported by the National Key R&D Program of China (2019YFD0900702) and the Agricultural Variety Improvement Project of Shandong (2019LZGC010); the National Key R&D Program of China (2019YFD0900702) and the Agricultural Variety Improvement Project of Shandong (2019LZGC010); and the Herbaceous Plant Germplasm Investigation Project of Shandong Province (No. SDGP370000000202202004592). The

funders had no role in study design, data collection and analysis, decision to publish, or preparation of the manuscript.

## Grant Disclosures

The following grant information was disclosed by the authors:

National Key R&D Program of China: 2019YFD0900702.

Agricultural Variety Improvement Project of Shandong: 2019LZGC010.

National Key R&D Program of China: 2019YFD0900702.

Agricultural Variety Improvement Project of Shandong: 2019LZGC010.

Herbaceous Plant Germplasm Investigation Project of Shandong Province: SDGP370000000202202004592.

## Competing Interests

The authors declare there are no competing interests.

## Author Contributions

- Shuang Li conceived and designed the experiments, performed the experiments, analyzed the data, prepared figures and/or tables, authored or reviewed drafts of the article, and approved the final draft.
- Yanling Yin conceived and designed the experiments, analyzed the data, prepared figures and/or tables, authored or reviewed drafts of the article, and approved the final draft.
- Jianmin Chen analyzed the data, authored or reviewed drafts of the article, and approved the final draft.
- Xinyu Cui analyzed the data, authored or reviewed drafts of the article, and approved the final draft.
- Jinmin Fu conceived and designed the experiments, analyzed the data, authored or reviewed drafts of the article, and approved the final draft.

## Data Availability

The raw measurements are available in the Supplementary File.

## Supplemental Information

Supplemental information for this article can be found online at http://dx.doi.org/10.7717/peerj.16985#supplemental-information.

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
