# Peer review of "H2O2 promotes trimming-induced tillering by regulating energy supply and redox status in bermudagrass"

_PeerJ, doi:10.7717/peerj.16985_

## Round 0.1 · original submission · Major Revisions

Although the manuscript shows a nice correlation between LOG1 and IPT gene expression and Zeatin levels upon trimming, as Reviewer 1 justly points out, this overlooks the role of CKX1, the main cytokinin degrading enzyme. It seems relatively straightforward to obtain and include the expression data for this gene to make the gene expression study more conclusive.

The second reviewer provides a list of more specific comments that should be considered in order to further improve the manuscript.

**Language Note:** The review process has identified that the English language must be improved. PeerJ can provide language editing services - please contact us at copyediting@peerj.com for pricing (be sure to provide your manuscript number and title). Alternatively, you should make your own arrangements to improve the language quality and provide details in your response letter. – PeerJ Staff

Reviewer 1 ·

Basic reporting

All my conclusions you can read in last window (#4)

Experimental design

All my conclusions you can read in last window (#4)

Validity of the findings

No comments

Additional comments

Dear authors,

It was evidenced, that cytokinins and auxins reduce the levels of H2O2 in guard cells and induced stomatal opening in darkness. Additionally, cytokinins not only reduced exogenous H2O2 levels in guard cells caused by exposure to light, but also abolished H2O2 that had been generated during a dark period, and promoted stomatal opening:
Song Xi-Gui, She Xiao-Ping, He Jun-Min, Huang Chen, Song Tu-sheng (2006) Cytokinin- and auxin-induced stomatal opening involves a decrease in levels of hydrogen peroxide in guard cells of Vicia faba. Functional Plant Biology 33, 573-583. https://doi.org/10.1071/FP05232
From other hand, tillering of cereals can be improved by cytokinins
Koprna, R.; Humplík, J.F.; Špíšek, Z.; Bryksová, M.; Zatloukal, M.; Mik, V.; Novák, O.; Nisler, J.; Doležal, K. Improvement of tillering and grain yield by application of cytokinin derivatives in wheat and barley. Agronomy 2021, 11, 67. https://doi.org/10.3390/ agronomy11010067
Therefore here is a big question about interplay of cytokinins and H2O2 in regulation of tillering after trimming/pruning/
The authors consider content of cytikinins during their experiments, paying attention to the synthesis of these phytohormones. But on our opinion the regulation of genes of cytokinin oxidase/dehydrogenase (CKX), encoding a cytokinin degrading enzyme could be more important under this angle view. This story can tracked from experiments with rice (Ashikari, M., Sakakibara, H., Lin, S., Yamamoto, T., Takasi, T., Nishimura, A., et al. (2005). Cytokinin oxidase regulates rice grain production. Science 309 (5735), 741–745. doi:10.1126/science.1113373 ), barley (Zalewski, W., Galuszka, P., Gasparis, S., Orczyk, W., Nadolska-Orczyk, A. (2010). Silencing of the HvCKX1 gene decreases the cytokinin oxidase/dehydrogenase level in barley and leads to higher plant productivity. J. Exp. Bot. 61, 1839–1851. doi:10.1093/jxb/erq052) and finger millet to last stories, which can be found in: https://www.frontiersin.org/research-topics/23523/harnessing-cytokinin-biology-in-crop-biofortification-and-enhanced-food-security#articles

I recommend authors to reconsider the ideology of the paper, involving in discussion mentioned above statements or to conduct out additional experiments on CKX gene expression.

Reviewer 2 ·

Basic reporting

no comment

Experimental design

no comment

Validity of the findings

no comment

Additional comments

This study investigated the morphological and physiological changes of bermudagrass after trimming treatment, and found that trimming can facilitate the development of tillers via promoting H2O2 burst. This study is very interesting, however, there are some questions should be addressed:
1. The title doesn't focus the article well, so please revise and improve it.
2. L102, please list the formula or reference of the Hoagland nutrient solution.
3. L122, 6-BA was raised for the first time, please complete its full name.
4. L136, L139, does the kit used remove genomic DNA?
5. L143, add references for Zeatin and sugar measurement methods.
6. L160, for plants, the response of new and old leaves to treatment is different. Therefore, it should be clearly described which part of the leaves were used to measure chlorophyll a fluorescence transient curve.
7. L163, the levels of H2O2 exogenous treatment…, please confirm the meaning of the expression.
8. L166, add references for starch content determination.
9. L194, ‘36 M-1 cm-1’should be changed into ‘36 M-1 cm-1’. I also can find the wrong formats in other text of the manuscript, please check and revise it.
10. L237-L240, the two paragraphs express the same thing and it is suggested that they be integrated in one paragraph.
11. L336-L357, this paragraph has a relatively confusing narrative logic and lacks relevant literature to support it, suggesting further discussion of it.
12. L370, ‘cytokinin’, the rest of the article uses CKs, suggesting change it.
13. L408, references are cited in different formats, make sure the formatting is standardized.
14. L519, ‘H2O2’ should be changed into ‘H2O2’.
15. The coordinate font size of Figure 8 is clearly different from the other graphs, please standardize icons.

---

## Round 0.2 · accepted · Accept

You have addressed all of the reviewers comments, therefore to my mind the manuscript is now ready for publication.

Reviewer 1 ·

Basic reporting

In principle, all my questions were answered.
For full picture revised part about the gene expression CKs degradation enzymes CKX could be supplemented by Radchuk et al. (doi: 10.1093/jxb/ers200) - only one paper about genome analysis of CKX expression linked not only with grain yield, but with tillering, too.

Experimental design

Good

Validity of the findings

Good

Additional comments

None